# Water and chloride as allosteric inhibitors in WNK kinase osmosensing

Liliana R Teixeira, Radha Akella, John M Humphreys, Haixia He,
Elizabeth J Goldsmith*

Department of Biophysics, The University of Texas Southwestern Medical Center,
Dallas, United States

## eLife Assessment

This study presents an **important** investigation of water coordination in a specific kinase family with a focus on the regulation of osmosensing protein kinases. X-ray crystallographic approaches combined with functional assays are used to address the hypothesis that bound water participates in the osmosensing mechanism as an allosteric kinase inhibitor. The evidence for changes in kinase conformation and space group of the crystal as a function of added low molecular weight polyethylene glycol is **solid**. The work will be of considerable interest to the kinase field as well as colleagues studying allosteric regulation of protein function.

*For correspondence:
elizabeth.goldsmith@
utsouthwestern.edu

**Competing interest:** The authors declare that no competing interests exist.

**Abstract** Osmotic stress and chloride regulate the autophosphorylation and activity of the WNK1 and WNK3 kinase domains. The kinase domain of unphosphorylated WNK1 (uWNK1) is an asymmetric dimer possessing water molecules conserved in multiple uWNK1 crystal structures. Conserved waters are present in two networks, referred to here as conserved water networks 1 and 2 (CWN1 and CWN2). Here, we show that PEG400 applied to crystals of dimeric uWNK1 induces de-dimerization. Both the WNK1 the water networks and the chloride-binding site are disrupted by PEG400. CWN1 is surrounded by a cluster of pan-WNK-conserved charged residues. Here, we mutagenized these charges in WNK3, a highly active WNK isoform kinase domain, and WNK1, the isoform best studied crystallographically. Mutation of E314 in the Activation Loop of WNK3 (WNK3/E314Q and WNK3/E314A, and the homologous WNK1/E388A) enhanced the rate of autophosphorylation, and reduced chloride sensitivity. Other WNK3 mutants reduced the rate of autophosphorylation activity coupled with greater chloride sensitivity than wild-type. The water and chloride regulation thus appear linked. The lower activity of some mutants may reflect effects on catalysis. Crystallography showed that activating mutants introduced conformational changes in similar parts of the structure to those induced by PEG400. WNK activating mutations and crystallography support a role for CWN1 in WNK inhibition consistent with water functioning as an allosteric ligand.

## Introduction

With No Lysine (WNK) kinases are soluble intracelluar serine–threonine kinases noted for their unique active site (*Xu et al., 2000*) and their association with familial hyperkalemic hypertension (FHHt) (*Wilson et al., 2001*). WNKs are the chloride-inhibited protein kinases long anticipated to control the activity of cation chloride cotransporters in response to osmotic stress (*Lytle and Forbush, 1992*). Previous data have implicated WNKs as homeostatic regulators of the intracellular milieu with respect to ions and osmotic pressure (reviewed in *Goldsmith and Rodan, 2023*; *Richardson and Alessi, 2008*; *Shekarabi et al., 2017*). WNKs are activated by osmotic stress in cells (*Xu et al., 2000*) and are inhibited by chloride (*Piala et al., 2014*). How chloride binds and inhibits the WNK1 kinase domain is

known (*Piala et al., 2014*), but how osmotic stress affects WNKs is only partially understood (*Akella et al., 2020*; *Akella et al., 2021*).

Osmotic pressure is a demand on solvent. Osmolytes are thought to mimic osmotic pressure by affecting waters of hydration on proteins (*Colombo et al., 1992*; *Leitner et al., 2020*; *LiCata and Allewell, 1997*; *Parsegian et al., 1994*; *Timasheff, 2002*; *Zhou et al., 2008*). WNKs have unique conserved conformation-specific cavities and corresponding water networks. WNKs de-dimerize and activate in osmolytes (*Akella et al., 2021*). A similar de-dimerization and activation occurs under hydrostatic pressure (*Humphreys et al., 2023*) suggesting the phenomena are related. Here we use crystallography to directly observe the effects of osmolytes on uWNK1 molecular structure, including de-dimerization and secondary structure changes. Mutagenesis of residues that line the largest water cavity, primarily using WNK3, probe the importance of the water network to WNK activity, regulation, and structure.

Previous crystallographic data have shown that the kinase domain of unphosphorylated WNK1 (WNK1/SA) adopts an asymmetric dimer configuration (*Akella et al., 2021*; *Min et al., 2004*). Small-angle x-ray scattering (SAXS) shows that the unphosphorylated kinase domain of WNK3 (uWNK3) adopts a similar configuration. Phosphorylated kinase domains of WNK1 and WNK3 (pWNK1 and pWNK3, 'p' for phosphorylated) are monomeric (*Akella et al., 2020*) (PDB files 5W7T and 5O26). Chloride ion is a pan-WNK inhibitor that binds uWNK1 (*Piala et al., 2014*). Osmolytes induced de-dimerization as observed by static light scattering (SLS) and SAXS in both uWNK1 and uWNK3 (*Akella et al., 2021*). A model for WNK regulation, then, is the presence of a dimer–monomer equilibrium: the dimer is inactive and binds inhibitory ligands; the monomer

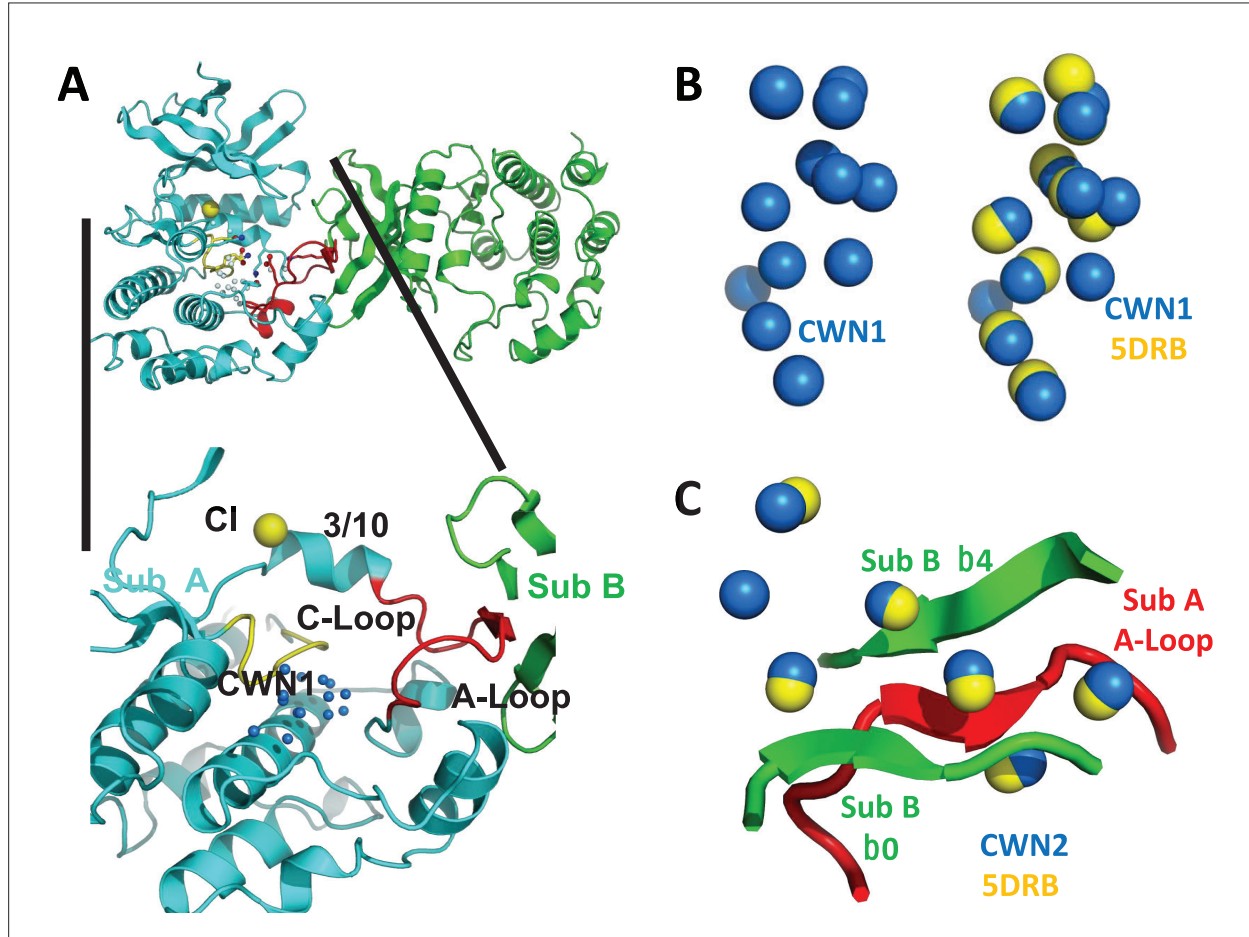

**Figure 1.** Conserved water networks in WNK1/SA. (**A**) Location of conserved water network 1 (CWN1) in Subunit A of uWNK/SA (PDB file 6CN9) shown as a closeup from the uWNK/SA dimer. CWN1 in marine, 14 water molecules. Subunit A, cyan, Subunit B, green, Activation Loop, red, and Catalytic Loop, yellow. (**B**) Conservation of waters in two crystal structures of uWNK1/SA, PDB 6CN9 (waters marine) and 5DRB (waters yellow). (**C**) Conserved water network 2 (CWN2) is in the subunit interface (same coloring as in (**A**)). N-terminal domains are superimposed.

is autophosphorylation-competent. We described previously that the inactive dimer uWNK1 has more bound water than the monomeric form (*Akella et al., 2021*). This is counter-intuitive, since the dimer should have a smaller surface area than the monomer. Instead, networks of water molecules present in cavities within the folded protein were independently observed in two uWNK1 structures (PDB files 6CN9 and 5DRB [*Yamada et al., 2016*]). The largest water network is conserved water network 1 (CWN1, *Figure 1A*). CWN1 is in the active site between the Catalytic Loop and the Activation Loop in uWNK1.

Here, we present crystallographic data that the osmolyte PEG400, when applied to crystals of uWNK1, induces de-dimerization and conformational changes. Comparison of the water structure in the inactive dimer versus the PEG400-induced conformation suggests that water functions as an inhibitor, binding and stabilizing an inactive WNK configuration. Furthermore, the osmolyte PEG400 favors a structure with less bound water.

Mutagenic analysis was performed on pan-WNK-conserved residues surrounding the water network CWN1 (*Akella et al., 2021*). Here, we analyzed the effect of mutations in CWN1-binding residues on WNK activity and regulation, anticipating that activating mutations will be found. Given the location of CWN1 in the active site, it is also anticipated that many mutants will negatively affect catalysis or substrate binding and be inhibitory. The mutagenic study was carried out primarily on WNK3, the kinase domain most active and most sensitive to osmotic pressure among WNK kinase domains tested. Although many mutants were inhibitory, activating mutations, WNK3/E314A and WNK3/E314Q were identified. The corresponding mutations were introduced in WNK1, exhibiting similar activation. WNK3/E314A crystallized and the conformational changes are described.

## Results

### Conserved water networks in WNKs

Available structures of unphosphorylated WNK kinase domains possess networks of bound water molecules. In contrast, phosphorylated active WNKs are monomerIc (*Akella et al., 2021*). ProBis-H2O (*Jukič et al., 2020*) was initially used to find water networks across WNK1 structures; then the networks were manually curated in PyMOL (see Methods). CWN1 was observed previously (*Akella et al., 2021*), and is in the active site between the Catalytic Loop and Activation Loop (*Figure 1A*). Of the 14 waters that make up CWN1 in PDB file 6CN9, nine are conserved in PDB 5DRB (*Figure 1B*). A second water network was uncovered, CWN2, found by superimposing the N-terminal domains of 6CN9 and 5DRB. CWN2 is in the subunit interface in 6CN9 (a lattice contact in 5DRB, *Figure 1C*). Of the seven CWN2 waters in 6CN9, six are conserved in 5DRB.

### PEG400 destabilizes and induces conformational changes in WNK1/SA

PEG400 is a strong osmolyte we have shown activates uWNK (*Akella et al., 2021*). PEG400 has a destabilizing effect both on WNK1/SA (*Figure 2A*) and WNK3/SA (not shown) as observed by differential scanning fluorimetry. To test the effects of PEG400 on WNK structure, we formed crystals and cryoprotected with either PEG400 or glycerol (as previously reported, see Methods).

Crystallographic data collection parameters and refinement statistics of the two structures are given in *Supplementary file 1a*. The structure observed in glycerol-soaked crystals was very similar to the previously determined uWNK1/SA structure 6CN9 (r.m.s.d. 0.19 Å), including a few glycerol molecules. No PEG400 was observed in uWNK1/SA/PEG400. Remarkably, PEG400 induced a space group change from P1 to P2$_1$. Together with the space group change, uWNK1/SA in PEG400 monomerizes. To understand the conformational change, the WNK1/SA/PEG400 structure was indexed in P1 and origin selected to place the 6CN9 dimer in the asymmetric unit (see *Supplementary file 1b*). The change in structure induced by PEG400 is shown in *Figure 2B*, giving an overall r.m.s.d. of 3.2 Å, between the 6CN9 dimer and the two subunits in WNK1/SA/PEG400. Superpositions of monomers show that the PEG400-induced monomer is more similar to the A-chain than the B-chain of 6CN9 (r.m.s.d of 0.79 versus 1.3 Å, *Supplementary file 1b*). *Figure 2C* gives the conformational change as a function of sequence. Prominent changes occur in helix C and β5, with the largest change occurring in the Activation Loop (including the preceding 3/10 chloride-binding helix). Changes in the same loci are induced by the osmolyte sucrose in phosphorylated WNK1 (*Akella et al., 2020*).

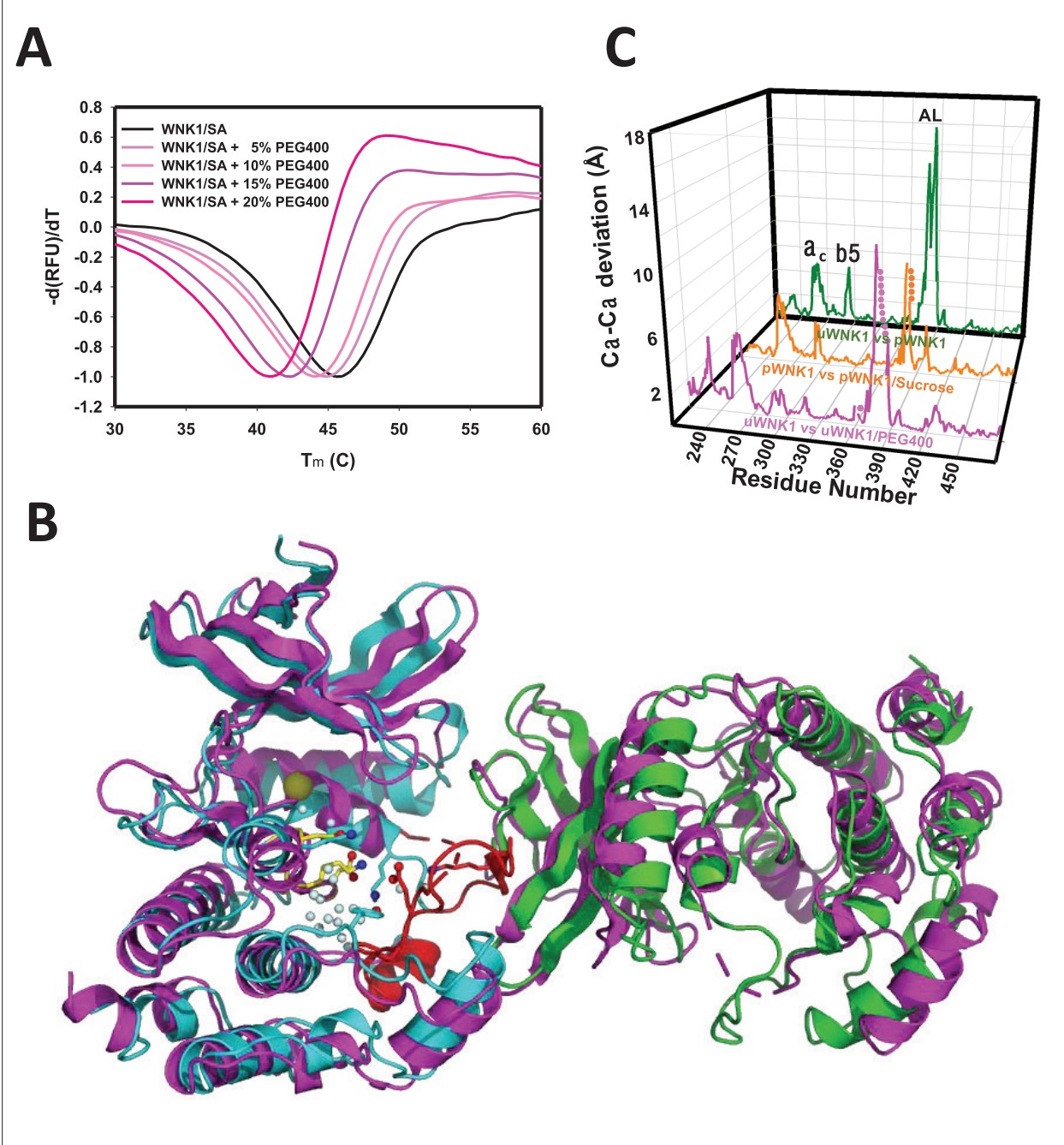

**Figure 2.** Effects of PEG400 on the WNK1/SA dimer and space group. (**A**) Differential scanning fluorimetry of WNK1 SA in increasing PEG400 (showing destabilization). (**B**) WNK1/SA/PEG400 (PDB 9D3F) has a single subunit in the asymmetric unit (magenta), shown with a symmetry mate (magenta) and overlayed with the asymmetric dimer of uWNK1/SA (PDB 6CN9, shown in cyan and green as in *Figure 1*). (**C**) Osmolyte-induced conformational changes as a function of sequence. WNK1/SA (PDB file 6CN9) versus WNK1/SA/PEG400, pink trace, pWNK1 (PDB 5W7T) versus pWNK1/sucrose (*Akella et al., 2020*), orange trace, and uWNK1 (6CN9) pWNK1 (5W7T), green trace.

The online version of this article includes the following source data for figure 2:

**Source data 1.** Differential scanning fluorimetry of WNK1/SA in PEG400 showing destabilization.

## PEG400 disrupts the water networks CWN1, CWN2, and the chloride-binding site

*Figure 3* is a closeup of the active site comparing the PEG400-induced structure with 6CN9. PEG400 affects CWN1, the chloride-binding 3/10 helix, and helix C (*Figure 3A*). PEG400 induces significant

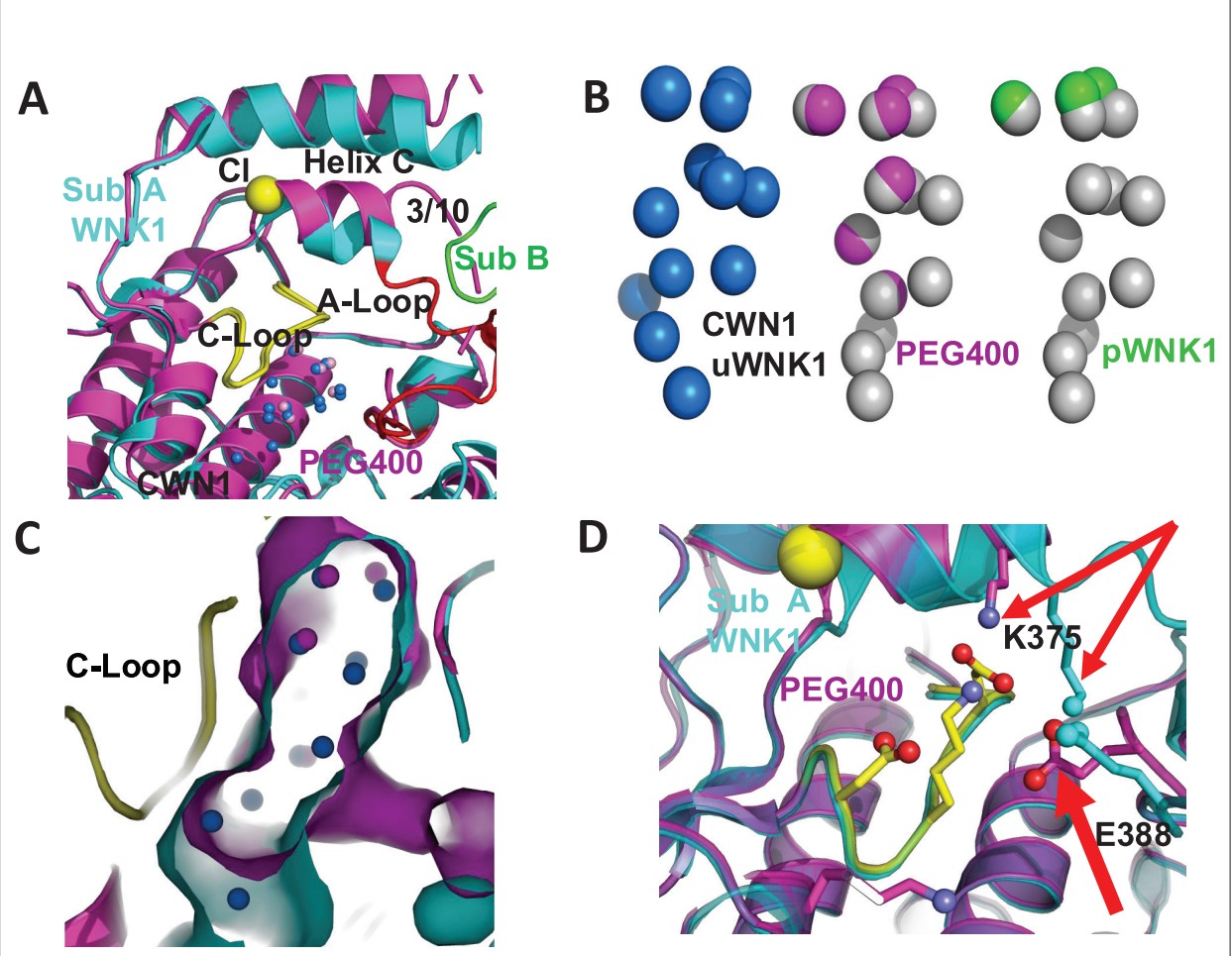

**Figure 3.** PEG400-induced conformational change in the active site of WNK1/SA. (**A**) PEG400-induced conformational changes in WNK1/SA. WNK1/SA Sub A (cyan), Sub B (green), superimposed with WNK1/SA/PEG400 (magenta). Conformational changes occur in the active site in both the 3/10 chloride-binding helix and the Activation Loop. (**B**) Comparison of conserved water network 1 (CWN1) in uWNK1 (blue), PEG400 (magenta), and in pWNK (lime). (**C**) Surface rendering in WNK1/SA/PEG400 showing the reduction in space near CWN. (**D**) Closeup of the Activation Loop highlighting residues E388 and K375 that move significantly in PEG400 (red arrows), affecting the 3/10 helix. Diagrams made in PyMOL.

losses in the CWN1, with only 5 waters remaining out of the 14 in 6CN9 (*Figure 3B*, missing waters in gray). An active configuration of WNK1, pWNK1 (PDB file 5W7T), has even fewer bound waters (*Figure 3B*). The loss of water is accompanied by a corresponding reduction in the cavity size (*Figure 3C*). Furthermore, PEG400 induces a large rotation in the 3/10 helix and neighboring helix C (*Figure 3D*). The change in the 3/10 helix eliminates the chloride-binding site and gives rise to big conformational changes at K375 and E388 in the Activation Loop (*Figure 3D*). Mutations at these sites are discussed below. The water network at the dimer interface, CWN2, is completely lost in PEG400 with the loss of the interface.

## CWN1 and the AL-CL Cluster

CWN1 in PDB 6CN9 is surrounded by a conserved cluster of charged residues. These residues are in the active site and emanate from the Activation Loop and Catalytic Loop (referred to here as the AL-CL Cluster) (*Figure 4A*). *Figure 4B* shows the high conservation of these residues across human WNK isoforms and WNKs from other species including *D. melanogaster*. Activation Loop residues E388, K381, and K375 of WNK1 interact with Catalytic Loop residues WNK1/D349, K351, and D353, respectively, forming a cage that houses CWN1 (*Figure 4A*). WNK1/D349 and WNK1/K351 are pan-kinase catalytic residues. Additional residues in the active site cavity, WNK1/K310, K375, K381, and Y420, contribute to the AL-CL Cluster. WNK1/E388 (WNK3/E314) is noteworthy because it is on the

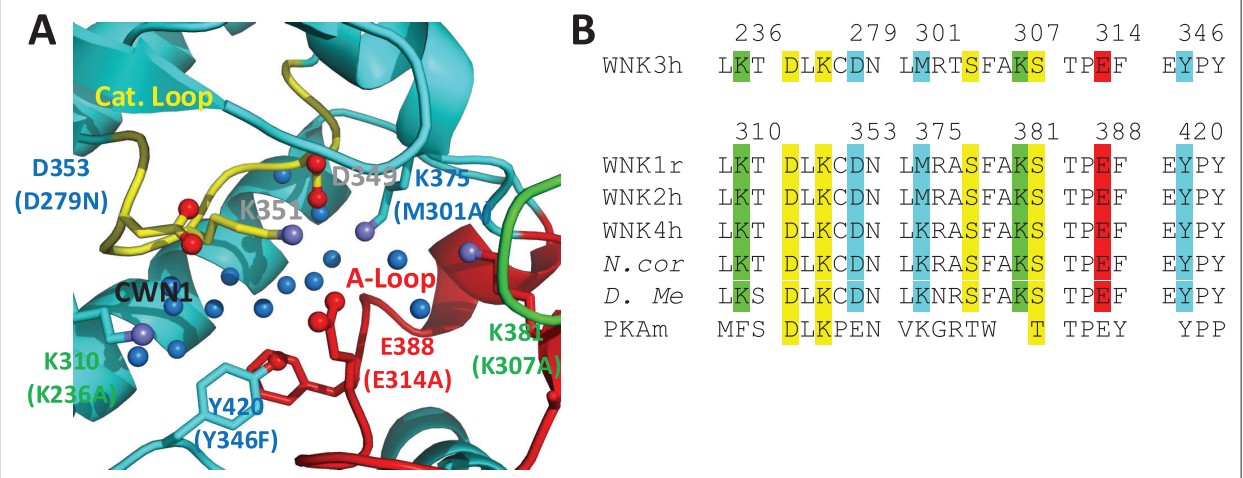

**Figure 4.** Residues in the AL-CL Cluster and positions mutated. (**A**) AL-CL Cluster in uWNK1 (PDB file 6CN9). Labels in WNK1 numbering (with WNK3 numbers in parenthesis). Cartoon coloring is the same as *Figure 1A*. Pan-kinase-conserved catalytic residues (D349 and K351) are labeled in gray. AL-CL residue labels are colored to indicate mutant assay results: mutants more active than wild-type, red, similar to wild-type, green, and less active than wild-type, blue. (**B**) Sequence conservation in the AL-CL Cluster and neighboring residues. Pan-kinase-conserved Catalytic Loop residues and pan-WNK Activation Loop phosphorylation sites are yellow, other colors indicate assay results, as in (**A**).

Activation Loop that undergoes large conformational changes upon phosphorylation (*Akella et al., 2020*).

## AL-CL Cluster mutant expression

We used mutational analysis to explore the role the AL-CL Cluster (and by proxy, CWN1), in WNK activity and regulation. Mutational analysis was carried out primarily in WNK3, the most active WNK in our hands. Activation Loop residues, WNK3/E314 (WNK1/E388), and K307 (WNK1/K381), and a residue in the Catalytic Loop, WNK3/D279 (WNK1/D353) were mutated. Additional AL-CL Cluster residues K236, M301 (lysine in WNK1), and Y346 were also mutated. AL-CL Cluster mutants were expressed in *Escherichia coli*, purified, and the state of phosphorylation assessed. Mutant expression levels varied, but all were sufficient for assays (*Supplementary file 1c*). Activation Loop phosphorylation at WNK3/S308 (WNK1/S382) and WNK3/S304 (WNK1/S378) is necessary for WNK activity (*Xu et al., 2002*). Chymotrypsin-derived peptides were monitored by liquid chromatography–mass spectrometry (LC–MS) to determine Activation Loop phosphorylation state (*Supplementary file 1d*). Each of the mutants was about 95% phosphorylated as expressed, suggestive of proper folding (*Supplementary file 1c,d*). Mutant proteins were dephosphorylated to form uWNKs. A range of autophosphorylation phenotypes (both more and less active than wild-type) was observed (*Figures 5 and 6*).

## Highly active WNK mutants

The higher activity mutants, WNK3/E314A and WNK3/E314Q are consistent with our model that CWN1 in uWNKs and the AL-CL Cluster that supports CWN1 are inhibitory. To determine whether mutation at this position is activating in other WNKs, we generated WNK1/E388A. As with WNK3, WNK1/E388A is more active than wtWNK1 (*Figure 5E, F*, *Supplementary file 1c*). (Also as expected, the basal autophosphorylation activity of wt uWNK1 is less than uWNK3, as discussed in the Introduction.) To determine whether chloride inhibition is retained in the higher activity mutants, the effects of chloride on uWNK3 (S308) and uWNK1 (S382) autophosphorylation were tracked. Under the conditions used, chloride is apparently less inhibitory of WNK3/E314A autophosphorylation (*Figure 5D*). Using WNK1/E388A, and in a slower reaction regime, the reduction in chloride inhibition is more obvious (*Figure 5E, F*). These data were fit to a simple autocatalytic kinetic model in Dynafit (*Kuzmič, 2009*) (see Methods). The modeling recapitulated the higher apparent activity and weaker chloride binding of WNK1 (*Figure 5E, F*, *Supplementary file 1f*). The high activity of the mutants WNK3/E314Q, WNK3/E314A, and WNK1/E388A suggests a loss of the CWN1-stabilized inhibitory conformation.

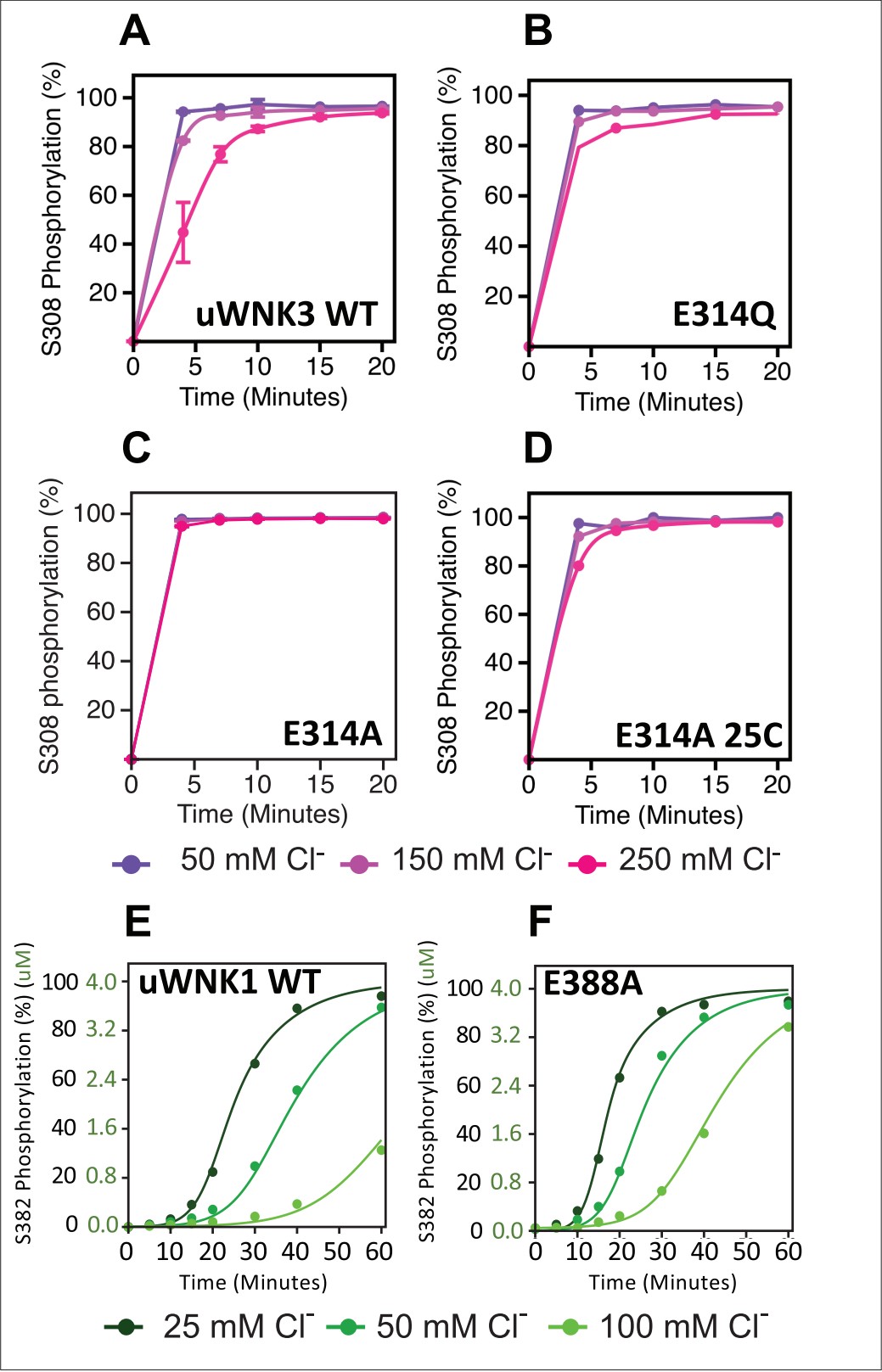

**Figure 5.** Basal autophosphorylation of WNK3/S308 and WNK1/S382 and effects of chloride on highly active mutations of position WNK3/E314 and WNK1/E388. (**A**) Wild-type uWNK3. (**B**) uWNK3/E314Q. (**C**) uWNK3/E314A. (**D**) uWNK3/E314A, 25°C. (**E**) uWNK1 activity and chloride inhibition. (**F**) uWNK1/E388A autophosphorylation activity. Lines in panels E and F are derived from DynaFit modeling and described in Methods and ***Supplementary***

*Figure 5 continued on next page*

*Figure 5 continued*

**file 1f**. Reactions run in 4 μM uWNK3, 30°C (unless otherwise indicated), at chloride concentrations of 50 mM (purple/dark green), 150 mM (pink/green), and 250 mM (magenta/light green). Bars indicate standard error from triplicate independent experiments.

The online version of this article includes the following source data and figure supplement(s) for figure 5:

**Source data 1.** Mass spectrometric quantitation of activation loop peptides encompassing WNK3/S308 or WNK1/S382 over time for wild-type and AL-CL Cluster mutants as a function of [Chloride].

**Figure supplement 1.** Effects of chloride on uWNK3 AL-CL Cluster mutant autophosphorylation at S304.

**Figure supplement 1—source data 1.** Mass spectrometric quantitation of WNK3/S304 phosphorylation over time in chloride.

## Low activity WNK mutants

*Figure 6* describes the activity and chloride sensitivity of wild-type uWNK3 (*Figure 6A*) and mutants that have either autophosphorylation activity similar to wild-type (*Figure 6B, C*); or lower activity than wild-type (*Figure 6D–F*). WNK3/K236 is on the periphery of the AL-CL Cluster and the mutant WNK3/K236A exhibits little change from wild-type. K307A also is very similar to wild-type, even though it

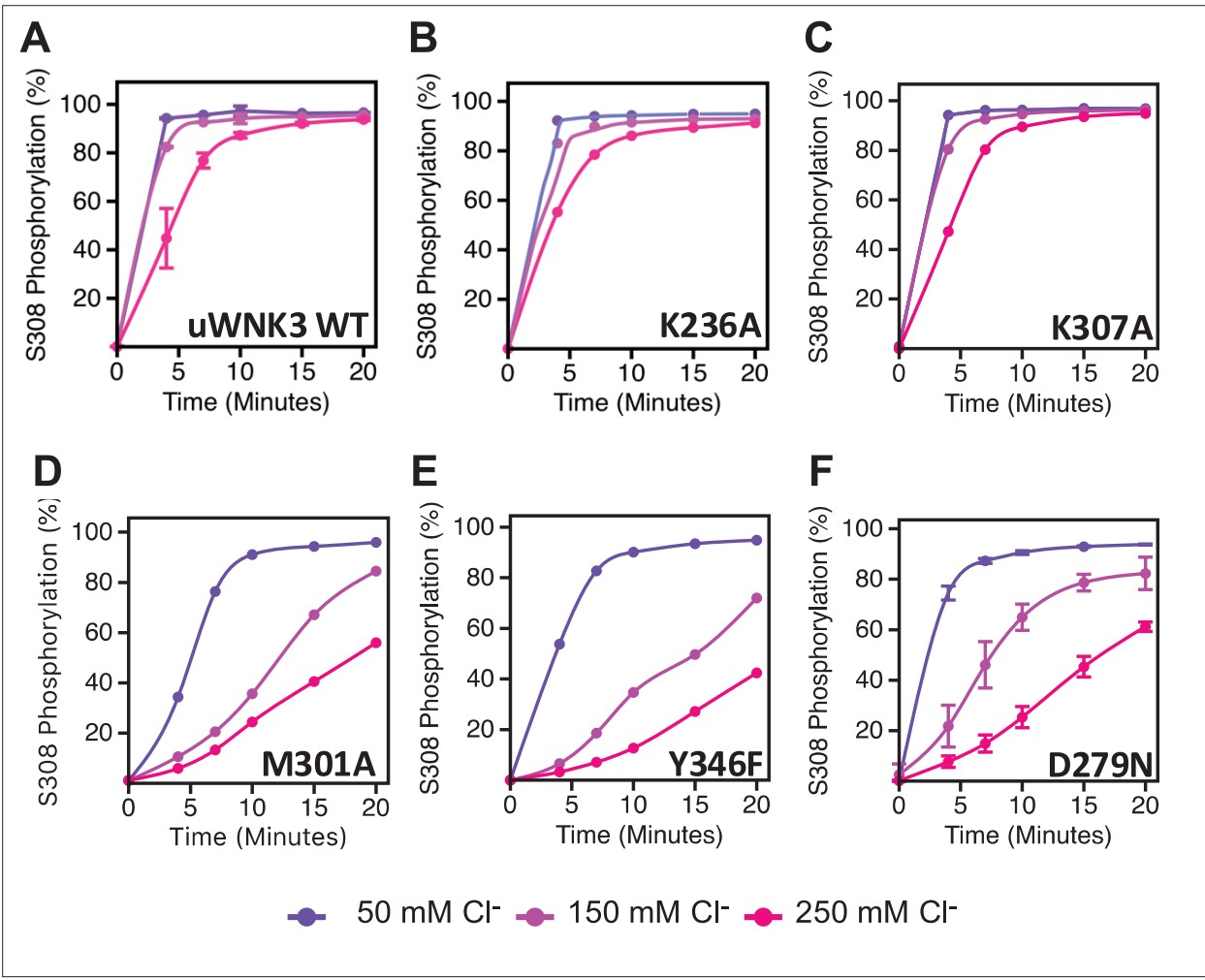

**Figure 6.** Basal autophosphorylation activity and effects of chloride on uWNK3 AL-CL Cluster mutants either similar to WT uWNK3 or less active. (**A**) Wild-type uWNK3. (**B**) uWNK3/K236A. (**C**) uWNK3/K307A. (**D**) uWNK3/M301A. (**E**) uWNK3/Y346F. (**F**) uWNK3/D279N. Reactions run in 4 μM uWNK3, 30°C at chloride concentrations indicated. S308 phosphorylation. Bars indicate standard error from triplicate independent experiments.

The online version of this article includes the following source data for figure 6:

**Source data 1.** Mass spectrometric quantitation of WNK3/S308 phosphorylation for less active mutants.

is adjacent to the phosphorylation site at Ser308. The mutants WNK3/D279N, M301A, and Y346F show related and complex phenotypes. These mutants are less active than WT and much more sensitive to chloride (*Figure 6D–F*). In addition to WNK3/S308, WNK3/S304 is also autophosphorylated (extent described in *Akella et al., 2021*). The effects of mutation on S304 phosphorylation mirrored the effects on S308 phosphorylation: wtWNK3 and WNK3/E314A were the most active, K236A and K307A were similar to wild-type, and M301A, Y346F, and D279N were the least active (*Figure 5—figure supplement 1*). S304 phosphorylation also appears to be more sensitive to chloride than S308 phosphorylation in the less active mutants (*Figure 5—figure supplement 1*).

## WNK3 AL-CL Cluster mutant osmotic sensitivity

The osmolyte PEG400 was used to put a demand on solvent in vitro as a mimic of osmotic pressure effects in cells (*Kuznetsova et al., 2014*). We previously observed that PEG400 stimulates ATP consumption by WNK3 in the presence of substrate GST-OSR1 (*Akella et al., 2021*). Unlike the LC–MS method used to study chloride inhibition by measuring Activation Loop autophoshorylation, ADP-Glo measures total ATP consumption and is compatible with PEG400.

We used ADP-Glo to measure GST-OSR1 substrate and autophosphorylation in all six WNK3 mutants (*Figure 7*). The ADP-Glo assay results are generally similar to the LC–MS autophosphorylation assays in that mutation of WNK3/E314 and WNK3/K236 show high activity whereas other AL-CL Cluster mutants are less active. The mutant WNK3/K307A showed lower activity in the ADP-Glo assays than in the MS-based autophosphorylation assays. Osmotic effects on WNKs measured in vitro with osmolytes are generally smaller than the chloride effects (*Akella et al., 2021*; *Piala et al., 2014*). None of the active single-point mutants were sufficient to eliminate the activating effect of osmolyte.

## Light scattering of mutant WNK3

SLS was performed on wild-type and mutant uWNK3 between 0.8 and 1.8 mg/ml at room temperature as described previously (*Akella et al., 2021*; *Supplementary file 1g*). At 0.8 mg/ml, wild-type uWNK3 exhibits an apparent mass of ~60 kD, suggestive of a 50/50 mixture of 40 kD monomer and 80 kD dimer. At 1.8 mg/ml uWNK3 is dimeric. In contrast, wild-type uWNK1 is monomeric at both concentrations. The WNK3 mutants generally exhibited 50/50 mixtures at 0.8 mg/ml, similar to wild-type uWNK3. Two of the mutants, WNK3/E314Q and WNK3/D279N were monomeric. The appearance of monomers on mutating the AL-CL Cluster support the idea that the waters in CWN1 promote the dimer.

## Crystal structure of uWNK3/E314A

To address whether conformational changes induced by the mutations correlate with the effects of PEG400, WNK3/SA/E314A and the corresponding WNK1/SA/E388A were crystalized. Crystals of WNK1/SA/E388A did not diffract. WNK3/SA/E314A crystallized, affording limited diffraction to 3.4 Å (see Materials and methods and *Supplementary file 1h*). WNK3/SA/E314A crystals were in space group P1, and adopts a unique dimer in which the Activation Loops are swapped. The significance of this domain swapping is unclear. Excluding the Activation Loops, comparisons of uWNK3/SA/E314A, uWNK1/SA, and uWNK1/SA/PEG400 show that uWNK3/SA/E314A is more similar to uWNK1/SA/PEG400 than uWNK1/SA (with r.m.s.d.'s of 0.7 versus 0.85, respectively). Furthermore, the orientations of the 3/10 helix and helix C most resemble WNK1/SA/PEG400 (*Figure 8A*). The greater similarity of uWNK3/SA/E314A to uWNK1/SA/PEG400 than to uWNK1/SA is apparent in the plot of conformational differences as a function of sequence (blue trace smaller than cyan trace, *Figure 8B*).

## Discussion

WNK kinases in cells are inhibited by chloride and activated by osmotic stress. WNK kinase domains exhibit multiple conformers. Inactive forms of both WNK1 (crystallography) and WNK3 (SAXS data) form asymmetric dimers. Prior data showed that chloride ion binds and promotes the inactive dimer, whereas osmolytes, in vitro surrogates for osmotic pressure, de-dimerize WNK kinase domains. Prior crystallographic data implicate a highly unique water-binding site, the AL-CL Cluster, in the inactive WNK1 dimer, as compared with typical protein kinases. Data presented here show that PEG400 applied to crystals of WNK1 rearranged and eliminated the AL-CL Cluster and the associated network

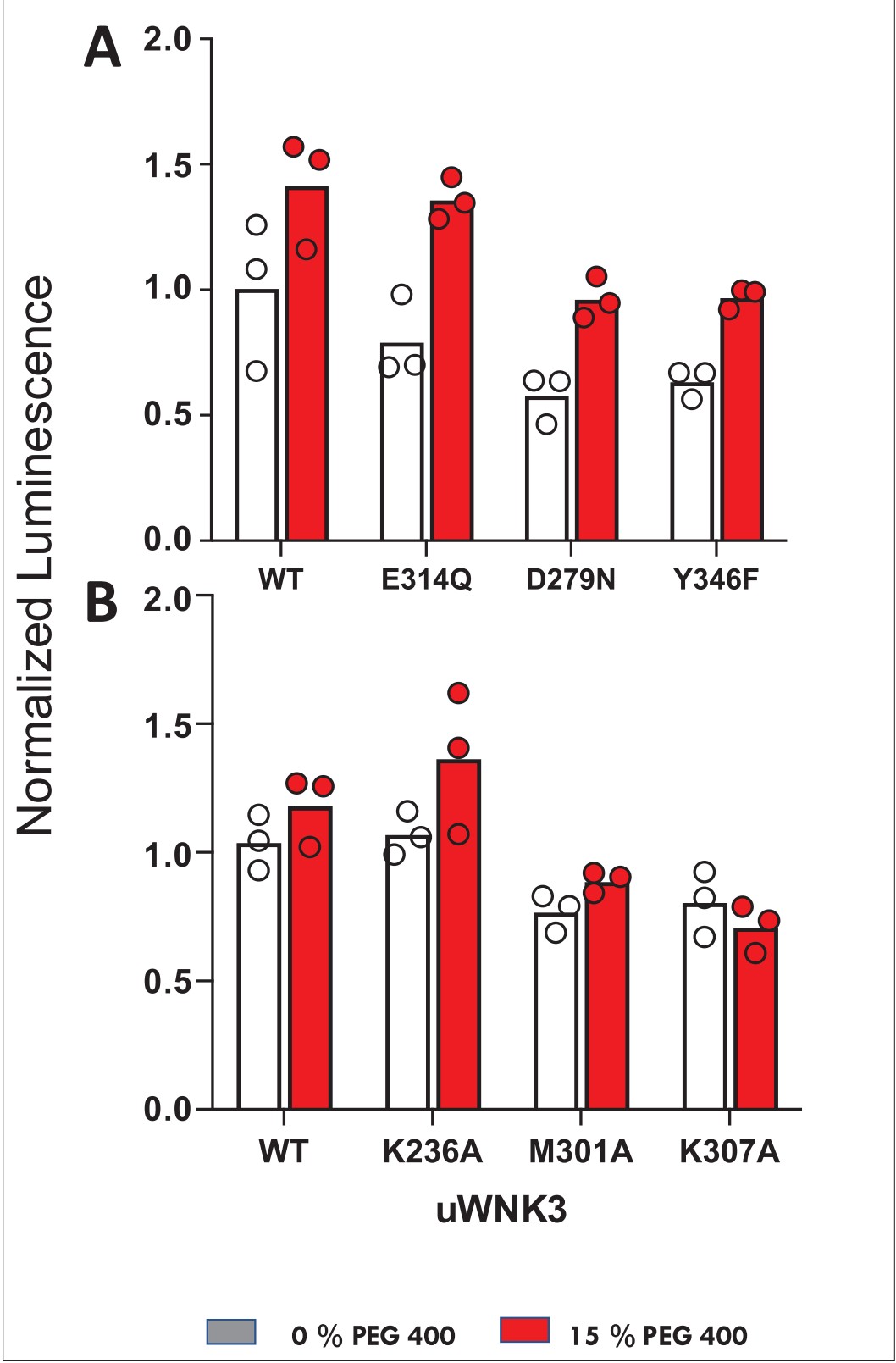

**Figure 7.** Effects of osmolytes on uWNK3 AL-CL Cluster mutant substrate phosphorylation. (**A**) WT-uWNK3, E314Q, D279N, and Y346F. (**B**) WT-uWNK3, K236A, M301A, and K307A. ADP Glo (Promega) with gOSR1 peptide as a substrate in the absence (gray) or presence (red) of 15% PEG400 15 min, 25°C.

*Figure 7 continued on next page*

*Figure 7 continued*

The online version of this article includes the following source data for figure 7:

**Source data 1.** Activity of WNK3 mutants toward gOSR1 as measured with ADP-Glo.

of water molecules, CWN1. PEG400 also eliminated the chloride-binding site and all the waters in CWN2. Apparently, the water networks can be allosteric ligands that promote the inactive structure of WNKs.

We probed the potential role of the water cluster CWN1 as an allosteric ligand by mutating residues in the AL-CL Cluster. Changes in autophosphorylation activity and chloride sensitivity were observed. Some AL-CL Cluster mutants, studied primarily in WNK3, were more monomeric than wild-type. Among the mutants assayed, WNK3/E314A,Q and WNK1/E388A exhibited enhanced activity. The enhanced activity of these mutants supports the hypothesis that the dimer is inhibitory and that CWN1 promotes the dimer. Crystallography of WNK3/E314A shows that it exhibits conformational differences at the same loci as observed by PEG400 applied to crystals of WNK1.

Several mutants were less active than wild-type. The mechanisms for the reduced activity are unclear. However, since the AL-CL Cluster and CWN1 are in the enzyme active site, reduced activity may be due to effects on substrate binding or catalysis.

Not all of the data presented here are easily interpreted in the framework of our inactive dimer-active monomer model. First, the activity and chloride sensitivity appear coupled in the AL-CL Cluster mutants. Mutants were either more active and less chloride sensitive, or less active and more chloride sensitive. The less chloride-sensitive mutants can be understood as arising from destabilization of the chloride-binding helix and the dimer. We have no interpretation so far for the enhanced chloride sensitivity of the low activity mutants. Second, ADP Glo assays, compatible with the crowding agent PEG400, showed a PEG-induced activation of substrate phosphorylation, consistent with our model. Among the active mutants, though, no single mutant abolished PEG sensitivity.

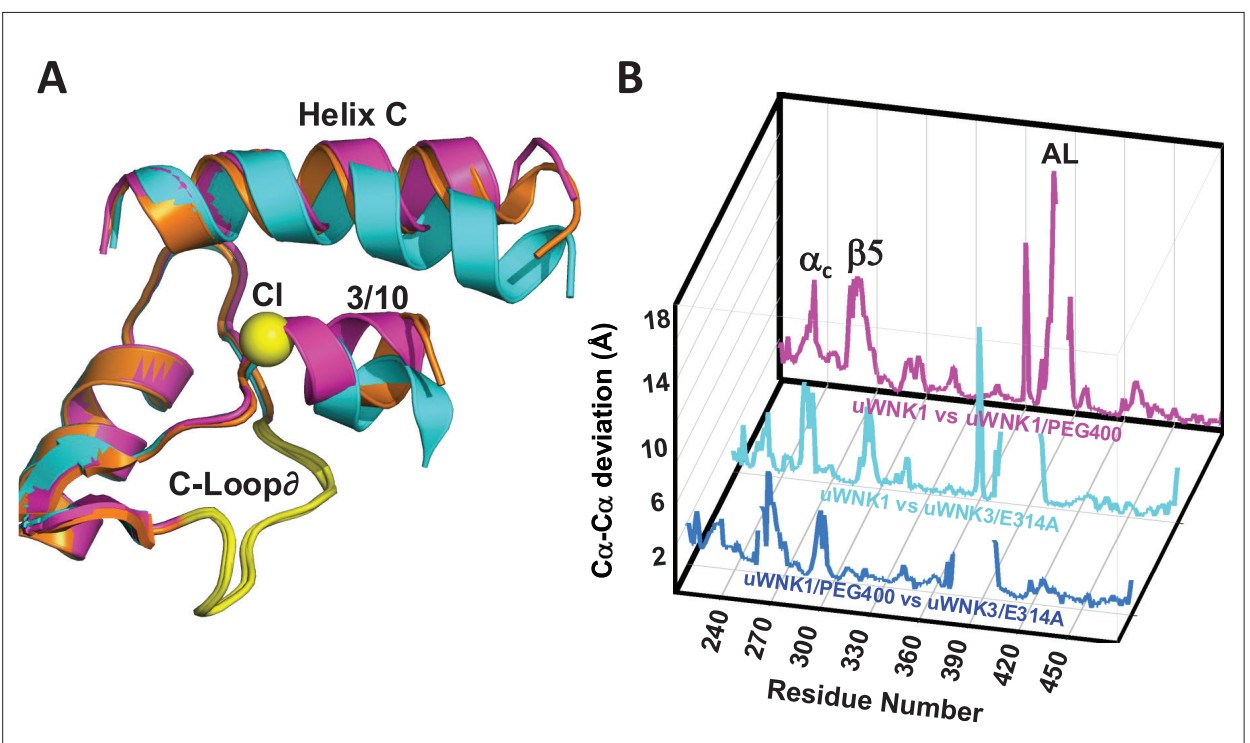

**Figure 8.** WNK3/SA/E314A structure overlay. (**A**) Colors the same as *Figure 3*, 6CN9 (cyan), uWNK1/SA/PEG400 (PDB file 9D3F, magenta), and WNK3/E314A (PDB file 9D7Q, orange). (**B**) Conformational differences as a function of sequence between uWNK1 and uWNK1/SA/PEG400 (pink), uWNK1 and uWNK3/SA/E314A (cyan), and uWNK1/SA/PEG400 and WNK3/SA/E314A (blue). Changes in Activation Loop residues have been omitted because they are off-scale due to disorder in PEG400 and domain-swapping in WNK3/E314A.

It has been posited recently that intracellular water potential is very tightly regulated in the face of osmotic and thermal stresses, and that water itself is the likely entity sensed to elicit responses (*Watson et al., 2023*). The present structural data support the idea that water is directly sensed in WNKs. We look forward to future biophysical characterization of other osmosensitive macromolecules to ascertain the generality of the themes observed here.

# Materials and methods

**Key resources table**

| Reagent type (species) or resource | Designation | Source or reference | Identifiers | Additional information |
|---|---|---|---|---|
| Recombinant DNA reagent | WNK1(194-483) (plasmid) | This paper | | GenScript |
| Recombinant DNA reagent | WNK1(194-483)_E388A (plasmid) | This paper | | GenScript |
| Recombinant DNA reagent | WNK3(118-409) (plasmid) | *Akella et al., 2021* | | GenScript |
| Recombinant DNA reagent | WNK3(118-409)_K236A (plasmid) | This paper | | GenScript |
| Recombinant DNA reagent | WNK3(118-409)_D279N (plasmid) | This paper | | GenScript |
| Recombinant DNA reagent | WNK3(118-409)_M301A (plasmid) | This paper | | GenScript |
| Recombinant DNA reagent | WNK3(118-409)_K307A (plasmid) | This paper | | GenScript |
| Recombinant DNA reagent | WNK3(118-409)_E314Q (plasmid) | This paper | | GenScript |
| Recombinant DNA reagent | WNK3(118-409)_E314A (plasmid) | This paper | | GenScript |
| Recombinant DNA reagent | WNK3(118-409)_Y346F (plasmid) | This paper | | GenScript |
| Recombinant DNA reagent | WNK3(118-409)_S308A_E314A (plasmid) | This paper | | GenScript |
| Recombinant DNA reagent | PP1cϒ (plasmid) | *Barford and Keller, 1994* | | From Depaoli-Roach |
| Peptide, recombinant protein | GST-OSR1(314-344) | *Taylor et al., 2018* | | From Melanie Cobb |
| Peptide, recombinant protein | Lambda phosphatase | Santa Cruz Biotechnology | Cat# sc-200312 | |
| Commercial assay or kit | ADP-Glo Max Assay | Promega | Cat# V7001 | |

An expression plasmid-encoding PP1cγ was a gift from Dr. Depaoli-Roach (Indiana University) and purified according to *Barford and Keller, 1994* as described previously. An expression plasmid-encoding GST-OSR (314-344) was obtained from Clinton Taylor IV and Melanie Cobb and purified as described in *Akella et al., 2021*. Standard reagents for purification and mass spectrometry we purchased from Sigma and Fisher Scientific. Lambda phosphatase was from Santa Cruz Biotechnologies. DNA for human WNK3 (118-409) and WNK1 (194-483) was subcloned into a pET29b vector by Genscript Inc (New Jersey) (*Akella et al., 2021*), leading to improved expression. Mutants of WNK3 and WNK1 based on the AL-CL Cluster in WNKs were also synthesized by Genescript, Inc. The mutants synthesized are WNK3/K236A, D279N, E314A, E314Q, K307A, M301A, Y346F, and WNK1/E388A (see Key resources table). WNK3/E314A was also inserted into a modified pET29b vector encoding a second TEV site between the S-tag at the N-terminus and the coding region (a gift from Aylin Rodan, Univ. of Utah and available on request). This expression vector was used to generate tag-free protein for crystallization.

## Expression and purification of WNK3, WNK1, and mutants

Purification of WNK1 kinase domain for crystallization was performed following protocols in *Min et al., 2004*. The kinase domain of WNK3 (WNK3(118-409)) expresses well in BL21(DE3) *E. coli* cells. WNK3 and mutants were expressed and purified in pET29b using modifications of the original protocol (*Min et al., 2004*). Colonies were grown in terrific broth (TB) containing the antibiotics kanamycin and chloramphenicol on an INFORS shaker overnight at 37°C. 800 µl of cultured cells were transferred to 50 ml of TB at 37°C and grown to $OD_{600}$ = 1.8; then 16 ml were transferred to 1 l of TB and grown to $OD_{600}$ = 1.3. Protein expression was induced with 0.5 mM isopropyl thiogalactopyranoside. Cultures were grown for 18 hr, at 18°C and 180 rpm. Cells were pelleted by centrifugation and flash frozen in liquid $N_2$. Cells were lysed in an Avastin cell disruptor, and the lysate ultracentrifuged. The WNK3 was extracted from the supernatant on a $NiSO_4$-charged Sepharose column (GE) which was loaded

in 50 mM Tris–HCl pH 8, 500 mM NaCl, 5 mM imidazole, and eluted in 250 mM imidazole. After dialysis in 50 mM Tris–HCl pH 8, 50 mM NaCl, 1 mM etheylenediamine tetra acetic acid (EDTA), and1 mM dithiothreitol, the WNK3s were further purified a Mono-Q 10/100 GL column (GE) eluted with 1 M NaCl. Samples were concentrated using an Amicon (Millipore), and buffer exchanged to 50 mM hydroxyethyl piperazine ethanesulfonic acid (HEPES) pH 7.4, 50 mM NaCl, 1 mM EDTA, and 1 mM Tris carboxyethyl phosphene. The HEPES buffer was made using HEPES free acid (Fisher) and adjusted to pH 7.4 using Tris base (Fisher). Gel filtration was performed on a Superdex-75 16/60 column. WNK1, WNK1/E388A used for assays were expressed and purified using similar protocol as WNK3.

WNK1/3 is phosphorylated (pWNK1/3) as purified from *E. coli*. The extent of phosphorylation of the different mutants is in *Supplementary file 1c*. Phosphorylated WNK1/3 (pWNK1/3) mutants were dephosphorylated using PP1c$\gamma$ and $\lambda$-phosphatase in a 10:1 ratio, in 0.5 mM MnCl$_2$. Phosphatases were removed using Ni-NTA and gel filtration chromatography as described (*Akella et al., 2021*). The uWNK1/3 was stored in 50 mM HEPES pH 7.4 and 150 mM NaCl.

## Crystallization and PEG400 treatment of uWNK1 crystals and x-ray methods

Crystals of WNK1/SA were obtained as reported previously (*Akella et al., 2021*; *Min et al., 2004*), grown in 24% PEG 2000, 300 mM NaCl, and 100 mM HEPES, pH 7.5, and cryoprotected in 15% glycerol, 26% PEG2000, and 300 mM NaCl, pH 7.5. Crystals were in space group P1, with cell constants $a = 38.30$ Å, $b = 57.8$ Å, $c = 65.7$ Å, $\alpha = 91.3°$, $\beta = 89.99°$, $\gamma = 90.89°$. Recently, we obtained crystals under different condition of reduced chloride containing 200 mM potassium formate and 20% vol/vol PEG3350. These crystals when cryoprotected with 15% glycerol diffracted to 2.0 Å. The unit cell dimensions of $a = 38.2$ Å, $b = 57.7$ Å, $c = 65.6$ Å, $\alpha = 89.0°$, $\beta = 89.6°$, $\gamma = 89.2°$ were similar (0.19 Å r.m.s.d.) to that reported earlier (PDB 6CN9). These crystals were then exposed to 25% PEG400 in the same buffer for 25 min to determine the effect of PEG400 on the crystals. The space group changed to P2$_1$ with altered lattice constants, $a = 38.3$ Å, $b = 56.8$ Å, $c = 65.3$ Å, $\beta = 95.4°$ without significant loss of resolution.

## Crystallization of uWNK3/E314A

The unphosphorylatable mutant uWNK3/S308A/E314A was crystallized using tag-free protein. Crystallization was carried out in 8% Tacsimate, pH 8.0 10% PEG3350 (Hampton Research). These crystals were cryoprotected in 25% glycerol. The space group was P2$_1$, and the cell dimensions were $a = 50.2$ Å, $b = 113.6$ Å, $c = 67.5$ Å, $\beta = 101.4°$.

## Generic assay conditions

Assays were conducted in 20 mM HEPES, pH 7.4, 20 mM MgCl$_2$, 5 mM ATP and usually in 150 mM NaCl 30°C. Reactions were initiated by the addition of uWNK3 or uWNK1 to 4 µM. Substrate phosphorylation assays contained 40 µM gOSR1. Reactions were stopped by addition of guanidine-HCl to a final concentration of 1 M. The content of the assay buffer is given in *Supplementary file 1e*.

## Autophosphorylation of uWNK3 and mutant uWNK3 methods

Chloride sensitivity of in vitro phosphorylation assays was caried out in a water bath at 30°C, in 20 mM HEPES pH 7.4, 20 mM Mg-gluconate and 5 mM ATP. Varying concentrations NaCl were added to the reaction mix. Chloride inhibition was measured at 50, 150, and 250 mM total Cl⁻. Reactions were started by adding 4 µM of uWNK3 or uWNK3 mutant. Aliquots of 50 µl were removed at time points: 0, 4, 10, 15, and 20 min and stopped by addition of guanidine-HCl to 1 M. Assay contents in *Supplementary file 1e*.

## Autophosphorylation of uWNK1 and uWNK1/E388A methods

Chloride sensitivity of in vitro autophosphorylation assays was carried out at 30°C, in 20 mM HEPES pH 7.4, 20 mM Mg-gluconate, and 5 mM ATP. Varying concentrations NaCl were added to the reaction mix. Chloride inhibition was measured at 50, 150, and 250 mM total Cl⁻. Reactions were started by adding 4 µM of uWNK1 or uWNK1/E388A. Aliquots of 50 µl were removed at time points: 0, 10, 20, 30, 40, and 60 min and stopped by addition of guanidine-HCl to a final concentration of 1 M.

## Mass spectrometry methods

The reaction mixes were diluted with chymotrypsin mix (rendering the solutions 0.5 µM chymotrypsin, pH 8.0, 500 mM guanidine-HCl, and 25 mM $CaCl_2$) and incubated overnight at 30°C. Peptide separation and detection by mass spectrometry was carried out as described previously (*Akella et al., 2021*). Briefly, high-performance liquid chromatography separation was conducted using an RP-C18 column (Phenomenex Aeris Widepore 150 × 2.1 mm) with a Shimadzu 10ADvp in line with a Thermo Finnigan LTQ. Phosphorylation ratios for WNK3/S308 and WNK3/S304 and WNK1/S382 were obtained by integrating the ion traces corresponding to *m/z* ranges for Activation Loop peptides. Because mutations were introduced into the Activation Loop, the phosphopeptides changed (see *Supplementary file 1d*).

## Modeling of autophosphorylation progress curves

Kinetic models for the phosphorylation reactions were defined in DynaFit (*Kuzmič, 2009*). DynaFit applies the Levenberg–Marquardt algorithm to perform non-linear least squares regression of progress curve data. Differential evolution of trial parameters is used to find a global minimum. A simple autocatalytic model (uWNK + pWNK -> pWNK + pWNK) was sufficient to capture the shape of the phosphorylation progress curves. Equations used and solutions of the ordinary differential equations are listed in *Supplementary file 1f*.

## ADP-Glo assay methods

ADP-Glo reagent (Promega Inc) was used as a readout for autophosphorylation as described previously (*Akella et al., 2021*). This reagent is compatible with the high ATP concentrations required for uWNK autophosphorylation assays, and can be used, unlike mass spectrometry, in the presence of PEG400. 50 µl reactions contained 40 mM HEPES (pH 7.4), 10 mM $MgCl_2$, 4 µM pWNK, and 40 µM gOSR1. Final chloride concentration was maintained at 150 mM. The reaction was started by the addition of 5.2 mM ATP. Reactions were stopped after 15 min with 50 µl of ADP-Glo reagent in the presence of 100 nM WNK463 pan WNK inhibitor (*Yamada et al., 2016*). Manufacturer's protocols were followed for the remaining steps of ATP depletion (40 min), conversion of ADP to ATP (1 hr). 100 µl aliquots were transferred to a 96-well plate and centrifuged for 2 min at 800 rpm. Luminescence was read on a CLARIOstar plate reader and data analyzed using MARS software (both reader and software, BMG Labtech, Ortenberg, GER). Data were further processed using GraphPad-Prism software.

## SLS methods

SLS was conducted on a Wyatt DynaPro Nanostar DLS, at 25°C. Samples of uWNK3 were prepared at four concentrations (0.8, 1.2, 1.8, and 2.4 mg/ml) in 50 mM HEPES pH 7.4 and 150 mM NaCl. Samples were centrifuged at 16,100 × *g* for 10 min to remove aggregates and particles, before 5 µl were loaded into a quartz cuvette. Light scattering at 90° was monitored until the detector voltage was stable, and then the scattering was monitored for ten times for 5 s, with three replicates. Data were analyzed using Dynamics version 7.5.0.17 (Wyatt Technologies).

## Comparing water structure across multiple PDB files

ProBis-H2O (*Jukič et al., 2017*) was used to find clusters of conserved water molecules in the nine WNK1 kinase structures deposited in the PDB. The results were manually curated in using PyMOL to compare WNK1 and WNK3 structures and to superpose domains locally.

## Crystal structure of WNK1/SA in PEG400

Crystals of WNK1/SA soaked with PEG400 (PEG-uWNK1) exhibited higher symmetry than the uWNK1 crystals (PDB 6CN9). The starting model for molecular replacement was the WNK1/SA (PDB 6CN9). The model was built in COOT based on the |2Fo − Fc| maps. Restrained refinement against 2.0 Å x-ray data of coordinates including 250 water molecules was conducted using TLS in REFMAC5 in CCP4. The final R-factor and R-free was 0.20 and 0.23, respectively (*Supplementary file 1a*).

## Crystal structure of uWNK3/E314A

WNK3/S308A/E314A crystals were cryoprotected with 25% glycerol. The x-ray data were collected at the Stanford synchrotron radiation light source beam line 12-2. Data collection and refinement

parameters for these mutants are given in *Supplementary file 1h*. WNK3/SA/E314A diffracted to 3.4 Å, with unit cell dimensions $a$ = 50.2 Å, $b$ = 113.6 Å, $c$ = 67.5 Å, $\alpha$ = 90.0°, $\beta$ = 101.4°, $\gamma$ = 90.0° and space group P2₁. The starting model for molecular replacement was the SW120619 inhibitor-bound WNK3/SA (PDB 8EDH). The model was built in COOT based on the |2Fo − Fc| maps. Final restrained refinement containing 117 water molecules against 3.3 Å x-ray data using REFMAC5 in the CCP4 suite for WNK3/SA/E314A and including TLS yielded an R-factor and R-free of 0.19 and 0.27, respectively.

## Stability measurements by differential scanning fluorimetry

Differential scanning fluorimetry was used to measure the protein melt temperature using the lipophilic dye (Sypro Orange) (*Pantoliano et al., 2001*). Stock solutions (1) 200 µM WNK1/SA (194-483) in 50 mM HEPES, 50 mM NaCl, pH 7.4, (2) buffer (50 mM HEPES, 150 mM NaCl, pH 7.4), (3) 100% PEG400, and (4) 0.2 µl of SyproOrange (labeled 2.5X, Life Technologies Inc) were mixed. Final 100 µl solutions were 5 µM WNK1 and varying concentrations of PEG400. 25 µL were added to 3 wells in a Bio-Rad Multiplate 96-well clear PCR plate. Plates were read on a Bio-Rad CFX96 Real-Time PCR system. The temperature was increased from 4 to 80°C in 0.5°C steps and fluorescence measurements taken in the 6-FAM Fluorescein channel. The fluorescence response to the heat curve was fit to a binding isotherm with the inflection point taken as the melt temperature $T_m$.

## Acknowledgements

Results shown in this report were derived from work performed at Argonne National Laboratory, Structural Biology Center (SBC) at the Advanced Photon Source. The SBC is operated by the U Chicago Argonne, LLC, for the U.S. Department of Energy, Office of Biological and Environmental Research under contract DE-AC02-06CH11357. Use of the Stanford Synchrotron Radiation Lightsource, SLAC National Accelerator Laboratory, is supported by the U.S. Department of Energy, Office of Science, Office of Basic Energy Sciences under Contract No. DE-AC02-76SF00515. The SSRL Structural Molecular Biology Program is supported by the DOE Office of Biological and Environmental Research, and by the National Institutes of Health, National Institute of General Medical Sciences (including P41GM103393). Crystallographic studies were coordinated by Diana Tomchick in the UT Southwestern Structural Biology Laboratory. We thank Chad A Brautigam and the UTSW Molecular Biophysics Resource Core facility for static light scattering data collection and analysis.This work was supported the Mary Kay Ash Foundation International Research Scholar Program (LRT), National Institutes of Health (DK110358 to EJG) the Welch Foundation grant I1128 and I-2100-20220331, CPRIT grant RP190421, and endowment from Patti Bell Brown to EJG.

## Additional information

### Funding

| Funder | Grant reference number | Author |
|---|---|---|
| Mary Kay Ash Foundation | Liliana R Teixeira | Liliana R Teixeira |
| National Institute of Diabetes and Digestive and Kidney Diseases | DK110358 | Radha Akella<br>John M Humphreys<br>Haixia He<br>Elizabeth J Goldsmith |
| Welch Foundation | I1128 | Elizabeth J Goldsmith |
| Welch Foundation | I-2100-20220331 | Liliana R Teixeira |
| Cancer Prevention and Research Institute of Texas | RP190421 | Radha Akella<br>John M Humphreys<br>Haixia He<br>Elizabeth J Goldsmith |
| Endowment from Patti Bell Brown | | Elizabeth J Goldsmith |

| Funder | Grant reference number | Author |
| --- | --- | --- |

The funders had no role in study design, data collection, and interpretation, or the decision to submit the work for publication.

## Author contributions

Liliana R Teixeira, Data curation, Formal analysis, Validation, Visualization, Methodology, Writing – original draft, Data acquisition; Radha Akella, Conceptualization, Data curation, Investigation, Methodology, Writing – review and editing; John M Humphreys, Data curation, Investigation, Methodology, Writing – review and editing; Haixia He, Data curation, Investigation; Elizabeth J Goldsmith, Conceptualization, Resources, Supervision, Investigation, Project administration, Writing – review and editing

## Author ORCIDs

Elizabeth J Goldsmith ⓘD https://orcid.org/0000-0001-8102-5012

Reviewer #1 (Public review): https://doi.org/10.7554/eLife.88224.3.sa1
Reviewer #2 (Public review): https://doi.org/10.7554/eLife.88224.3.sa2
Author response https://doi.org/10.7554/eLife.88224.3.sa3

# Additional files

## Supplementary files

• Supplementary file 1. Tables. (a) Crystallographic data and refinement of WNK1/S382A and WNK1/SA/PEG400. (b) Cell constant superposition comparisons. (c) WNK3/1 expression level, and Activation Loop S308/S382 phosphorylation and activity. (d) WNK1 and WNK3 peptides monitored by liquid chromatography–mass spectrometry (LC–MS). (e) uWNK1 and uWNK3 autophosphorylation assay conditions. (f) Wild-type and E388A uWNK1 kinetic model. (g) Molecular weight versus [WNK] by static light scattering. (h) Crystallographic data and refinement of WNK3/SA/E314A.

• GoldsmithEJMDAR checklist

## Data availability

Crystallographic data have been submiited to the PDB. 9D3F is WNK1+PEG400 and 9D7Q is WNK3/E314A. Source data has been provided for Figures 2A, 5, 6, 7, and Figure 5—figure supplement 1.

The following datasets were generated:

| Author(s) | Year | Dataset title | Dataset URL | Database and Identifier |
| --- | --- | --- | --- | --- |
| Akella R | 2024 | The structure of the kinase domain of WNK1 soaked in PEG400 | https://www.rcsb.org/structure/9D3F | RCSB Protein Data Bank, 9D3F |
| Akella R, Goldsmith EJ | 2024 | The structure of the kinase domain of WNK3/E314A | https://www.rcsb.org/structure/9D7Q | RCSB Protein Data Bank, 9D7Q |

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
