## [Editor Report · eLife Assessment]

This study presents an **important** investigation of water coordination in a specific kinase family with a focus on the regulation of osmosensing protein kinases. X-ray crystallographic approaches combined with functional assays are used to address the hypothesis that bound water participates in the osmosensing mechanism as an allosteric kinase inhibitor. The evidence for changes in kinase conformation and space group of the crystal as a function of added low molecular weight polyethylene glycol is **solid**. The work will be of considerable interest to the kinase field as well as colleagues studying allosteric regulation of protein function.

---

## [Referee Report · Reviewer #1 (Public review)]

This manuscript addresses the regulation of the osmosensing protein kinases, WNK1 and WNK3. Prior work by the authors has shown that these enzymes are activated by PEG400 or ethylene glycol and inhibited by chloride ion, and that activation is associated with a conformational transition from dimer to monomer. In X-ray structures of the WNK1/SA inactive dimer, a water-mediated hydrogen bond network was observed between the catalytic loop (CL) and the activation loop (AL), named CWN1. This led to the proposal that bound water may be part of the osmosensing mechanism.

The current study carries this work further, by applying PEG400 to Xtals of dimeric WNK1/SA. This results in a change in kinase conformation and space group, along with 4-9 fewer waters in CWN1 and the complete disappearance of another water cluster (CWN2) located at the dimer interface. Six conserved residues lining the CWN1 pocket in WNK3 are mutated to determine effects on activity and inhibition by chloride ion (measured by AL autophosphorylation) and monomer-dimer interconversion (light scattering).

The results show that two mutants (E314Q/A in WNK3) at a site central to the water cluster result in increased kinase activity (autophosphorylation), and increased SLS, interpreted as aggregation. Three sites (D279A, Y346F, M301A) inhibit kinase activity with varying effects on oligomerization - Y346A and M301A retain monomer-dimer ratios similar to WT while D279N promotes aggregation. K236A and K307A show activity and monomer:dimer ratios similar to WT. Selected mutants (E314Q, D279N, Y346F) and WT appear to retain osmosensitivity with comparable activation by PEG400.

The study concludes that osmolytes may activate the kinase by removing waters from the CWN1 and CWN2 clusters, suggesting that waters might be considered allosteric ligands that promote the inactive structure of WNKs. The differing effects of mutations may be ascribed to disruption of the water networks as well as inhibitory perturbations at the active site.

Comments on latest version:

The revised manuscript incorporated new experiments that satisfactorily addressed my concerns.

---

## [Referee Report · Reviewer #2 (Public review)]

This work tests the hypothesis that water coordination in WNK kinases is linked to allosteric control of activity. It is proposed that dimeric WNK is inactive and bound to some conserved water molecules, and that monomerization/activation involves departure of these waters. New data here include a crystal structure of monomeric WNK1 which shows missing waters compared to the dimeric structure, in support of the hypothesis. Mutant proteins of a different isozyme (WNK3) designed to disrupt water coordination were produced, and activity and quaternary structure were measured.

Comments on latest version:

The authors have largely addressed my concerns by making sure collection of mutants analyzed for autophosphorylation in Figure 6 are consistent with the measurement of osmotic sensitivity in Figure 7. The other changes in response to reviews have made a stronger manuscript in my opinion.

---

## [Author Response]

The following is the authors’ response to the original reviews.

**Reviewer #1 (Public Review):**
Weaknesses:Given that all mutants tested showed the same degree of activation by PEG400, it seemed possible that PEG400 might be an allosteric activator of WNK1/3 through direct binding interactions. Perhaps PEG400 eliminates CWN1/2 waters by inducing conformational changes so that water loss is an effect not a cause of activation. To address this it would be helpful to comment on whether new electron densities appeared in the X-ray structure of WNK1/SA/PEG400 that might reflect PEG400 interactions with chains A or B.

We re-evaluated the WNK1/SA/PEG400 electron density looking for non-protein densities larger than water. No new densities were found. However, we do observe a PEG400-destabilizing effect using differential scanning fluorimetry, and have included this data into Figure 2. We conclude that the effects on the water structure and destabilization are due to demands on solvent.

We have included in the second paragraph of the introduction references to primary literature that advance similar arguments to explain osmolyte induced effects on activity.

Specifically, Colombo MF, Rau DC, Parsegian VA (1992) Protein solvation in allosteric regulation: a water effect on hemoglobin. *Science* 256: 655-659 and LiCata VJ, Allewell NM (1997) Functionally linked hydration changes in *Escherichia coli* aspartate transcarbamylase and its catalytic subunit. *Biochemistry* 36: 10161—10167.

It would also be helpful to discuss any experiments that might have been done in previous work to examine the direct binding of glycerol and other osmolytes to WNKs.

We did not observe PEG400 in WNK1/SA/PEG400 despite effects on the space group and subunit packing. On the other hand, glycerol was observed in WNK1/SA, which was cryoprotected in glycerol (PDB file 6CN9). We have highlighted these differences in the second section of the results. A thorough analysis on the effects of various osmolytes on WNK structure, stability, and activity is a potential future direction.

The study would benefit from a deeper discussion about how to reconcile the different effects of mutations. For example, wouldn't most or all of the mutations be expected to disrupt the water network, and relieve the proposed autoinhibition? This seemed especially true for some of the residues, like Y420(Y346), D353(D279), and K310(K236), which based on Fig 3 appeared to interact with waters that were removed by PEG400.

The manuscript has been updated with new data and better discussion of this point. Given the inconsistencies on the effects of mutation in static light scattering (SLS), we addressed the possibility that the reducing agent was not constant across experiments. In a repeated study, including reducing agent (1 mM TCEP), we obtained results on mutant mass more similar to wild-type than in the original experiment. An exception was that two of the mutants were much more monomeric than wild-type. It follows that the network CWN1 stabilizes the inactive dimer. The reduced activity of some of the mutants probably reflects the position of CWN1 and the AL-CL Cluster in the active site, such that mutants can affect substrate binding or catalysis. This is now better discussed both in the data and discussion sections.

Mutants have a tendency to have complex effects on activity and structure. It was satisfying to find any activating mutants. We point out that we have been careful to present all of our data including mutants that are not easily explained by our models.

Alternatively, perhaps the waters in CWN2 are more important for maintaining the autoinhibited structure. This possibility would be useful to discuss, and perhaps comment on what may be known about the energetic contributions of bound water towards stabilizing dimers.

This research focused on the most salient unique feature of WNK1- CWN1. We also identified CWN2. Mutational analysis of CWN2 can’t be done without disrupting the dimer interface, greatly complicating data interpretation.

It would also be useful to comment on why aggregation of E319Q/A (E314) shouldn't inhibit kinase activity instead of activating it.

On recollection of the SLS data in the presence of reducing agent, we saw reduced aggregation. WNK3/D279N and WNK3/E314Q were more monomeric, especially at the higher protein concentration used. WNK3/E314Q is one of the more active mutants.

The X-ray work was done entirely with WNK1 while the mutational work was done entirely with WNK3. Therefore, a simple explanation for the disconnect between structure and mutations might be that WNK1 and WNK3 differ enough that predictions from the structure of one are not applicable to mutations of the other. It would be helpful to describe past work comparing the structure and regulation of WNK1 and WNK3 that support the assumption of their interchangeability.

We have responded directly to this concern. We introduced our most interesting amino acid replacement WNK3/E314A into WNK1, making WNK1/E388A. Similar trends in chloride inhibition and mutational activation were observed in WNK1 as in WNK3. This supports the assumption of interchangeability of WNK1 and WNK3 we invoked for practical reasons. As expected, the overall activity of WNK1 is lower than WNK3. Overall, the lower activity limited data collection. However, the lower activity did allow us to fit the chloride inhibition data to a kinetic model for WNK1. Panels on WNK1 activity, mutation, and chloride inhibition were added to Figure 5 and to Supplemental data (Table S6).

**Reviewer #2 (Public Review):**
Strengths:The most interesting result presented here is that P1 crystals of WNK1 convert to P21 in the presence of PEG400 and still diffract (rather than being destroyed as the crystal contacts change, as one would expect). All of the assays for activity and osmolyte sensing are carried out well.

Thank you. We have emphasized this point in the Results section with the word “remarkably”

Weaknesses:The rationale for using WNK3 for the mutagenesis study is that it is more sensitive to osmotic pressure than WNK1. I think that WNK1 would have been a better platform because of the direct correlation to the structural work leading to the hypothesis being tested. All of the crystallographic work is WNK1; it is not logical to jump to WNK3 without other practical considerations.

This point is addressed in the last comment to Reviewer 1. We added autophosphorylation assay data on our most interesting mutant (WNK3/E314A) in WNK1 (WNK1/E388A). Conversely, we have crystallographic data on uWNK3 (on uWNK3/E314A collected to 3.3Å). These new data justify the assumption of interchangeability of results obtained for uWNK1 and uWNK3.

Osmolyte sensing was tested by measuring ATP consumption as a function of PEG400 (Figure 6). Data for the subset of mutants analyzed by this assay showed increasing activity. It is not clear why the same collection of mutant proteins analyzed in the experiments of Figure 5 was not also measured for osmolyte sensing in Figure 6.

These data are now more complete, having been now collected for all of the WNK3 mutants (now Figure 7).

The last set of data presented uses light scattering to test whether the WNK3 mutant proteins exhibit quaternary structural changes consistent with the monomer/dimer hypothesis. If they did, one would expect a higher degree of monomer for those that are activated by mutation, and a lower amount of monomer (like wt) for those that are not. Instead, one of the mutant proteins that showed the most chloride inhibition (Y346F) had a quaternary structure similar to the wt protein, and others have similar monomer/dimer mixtures but distinct chloride inhibition profiles (K307A and M301A). I don't see how the light scattering data contribute to this story other than to refute the hypothesis by showing a lack of correlation between quaternary structure, water binding, and activity. This is another reason why the disconnect between WNK1 and WNK3 could be a problem. All of the detailed structural work with WNK1 must be assumed with WNK3; perhaps the light scattering data are contradicting this assumption?

As noted above, on recollection of the SLS data in the presence of reducing agent, we saw reduced aggregation and more consistency with our model. Thus, we now feel it is a useful contribution to the manuscript. The table in Supplemental data has been updated.

**Reviewer #1 (Recommendations For The Authors):**
Fig 3D in the PDF manuscript seemed distorted - waters were cut off. Also Fig 2D would benefit from showing the whole molecule, instead of cutting off the top and bottom of the kinase domain.We suspect this is a data transfer problem, since we don’t see these truncations.

Both Figure 2 and 3 have been changed, addressing these concerns and adding new differential scanning fluorimetry data as discussed in reply to Reviewer 1. Figure 2 was simplified by eliminating Figures 2A-2C, and replacing them with a new Figure 2B, the superposition of WNK1/SA/PEG400 (PDB 9D3F), WNK1/SA (PDB 6CN9).

In Figure 3, we added a panel highlighting the volume change around CWN1 in presence of PEG400 (Figure 3C). Hopefully, inappropriate cropping has been eliminated.

Line 162: Y314F should be Y346F.

This has been corrected. Thank you.

Lines 211-213 - these two sentences do not seem to logically go together: "Two hyper-active mutants were discovered, WNK3/E314A, and WNK3/E314Q. These mutants are straightforward to interpret based on our model: the mutated residues support and stabilize inactive dimeric WNK."

An extensive rewrite has been conducted to address the difference in activity between the higher activity mutants versus less active mutants, now discussed in two paragraphs, and two Figures, Figure 5 and 6. The SLS data, recollected with more reducing agent, has given more consistent results (Supplemental), making the discussion more straightforward (discussed above).

**Reviewer #2 (Recommendations For The Authors)**
I think WNK1 would be a better platform for mutagenesis than WNK3. Or minimally the authors should better justify the switch to WNK3 from WNK1. Analyze the same set of mutants in Figure 5 into Figure 6.

Again, we have added assay data on uWNK1/E388A, and structural data on uWNK3/E314A.

I would analyze the same set of mutants in Figures 5 and 6.

We have analyzed all of the WNK3 mutants in the ADP-Glo assays (Figure 7).

Will the P21 crystal form grow independently in PEG400?

Attempts to crystallize WNK1/SA or WNK3/SA or other constructs in PEG400 have been unsuccessful.

I would also add some context about the role of water in allosteric mechanisms. I know there is a long history in hemoglobin in which specific waters have been associated with the T and R states such as that by Marcio Colombo. There is a relatively recent article in J. Phys Chem. that would provide good context. Leitner et al., J. Chem. Phys. 152, 240901 (2020)

Thank you. Good call.